# Mineralocorticoid Receptor Signaling in Peripheral Blood Cells in Patients with Multiple Sclerosis

**DOI:** 10.3390/ijms25168883

**Published:** 2024-08-15

**Authors:** Franziska Küstermann, Kathy Busse, Johannes Orthgieß, Muriel Stoppe, Sarah Haars, Florian Then Bergh

**Affiliations:** 1Klinik und Poliklinik für Neurologie, University of Leipzig, 04103 Leipzig, Germany; franziska.kuestermann@medizin.uni-leipzig.de (F.K.); kathy.busse@vetmed.uni-leipzig.de (K.B.); or johannes.orthgiess@skhal.sms.sachsen.de (J.O.);; 2Faculty of Veterinary Medicine, University of Leipzig, 04103 Leipzig, Germany

**Keywords:** mineralocorticoid receptor, aldosterone, target gene, transcriptional regulation, multiple sclerosis

## Abstract

Multiple sclerosis (MS) is associated with alterations in neuroendocrine function, primarily the hypothalamic–pituitary–adrenal axis, including lower expression of the glucocorticoid receptor (GR) and its target genes in peripheral blood mononuclear cells (PBMC) or full blood. We previously found reduced mineralocorticoid receptor (MR) expression in MS patients’ peripheral blood. MS is being treated with a widening variety of disease-modifying treatments (DMT), some of which have similar efficacy but different mechanisms of action; body-fluid biomarkers to support the choice of the optimal initial DMT and/or to indicate an unsatisfactory response before clinical activity are unavailable. Using cell culture of volunteers’ PBMCs and subsequent gene expression analysis (microarray and qPCR validation), we identified the mRNA expression of OTUD1 to represent MR signaling. The MR and MR target gene expression levels were then measured in full blood samples. In 119 MS (or CIS) patients, the expression of both MR and OTUD1 was lower than in 42 controls. The expression pattern was related to treatment, with the MR expression being particularly low in patients treated with fingolimod. While MR signaling may be involved in the therapeutic effects of some disease-modifying treatments, MR and OTUD1 expression can complement the neuroendocrine assessment of MS disease course. If confirmed, such assessment may support clinical decision-making.

## 1. Introduction

Multiple sclerosis (MS) is the most common non-traumatic neurological disease that causes permanent disability in young adults, more often in young females [1]. It is a chronic inflammatory autoimmune disease, characterized by demyelination, axonal injury and gliosis. Although the etiology of MS remains unclear, the factors involved in its pathogenesis include a genetic predisposition, environmental influences and resulting alterations in the immune and neuroendocrine systems [2,3,4]. The key effector mechanism is a predominantly T-cell-dependent autoimmune process [5,6,7]. The main intrinsic neuroendocrine influence on the immune system is the hypothalamic–pituitary–adrenal axis (HPA axis). Searching for factors that may explain the dysregulation of the immune system leading to autoimmunity, we and others have studied HPA axis activity in patients with MS and shown that chronic or episodic HPA axis hyperactivity correlates with the disease course and progression [8,9,10]. One feature of this dysregulation is the diminished expression or/and function of the glucocorticoid receptor (GR) in peripheral blood mononuclear cells (PBMC) [11,12,13], limiting the anti-inflammatory effects of endogenous or therapeutic glucocorticoids. The degree of GR expression and its target gene induction has, in fact, recently been shown to be associated with the clinical response to i.v. steroids as a relapse treatment [14]. This observation introduces the perspective of using neuroendocrine assessments as a biomarker for some aspects of the disease course.

The downregulation of GR expression was accompanied by the downregulation of the closely related mineralocorticoid receptor (MR) in our previous study [15]. It is currently difficult to estimate the functional impact of this finding in MS, but a possible role of MR in the cause of autoimmunity has emerged. For instance, in mice, the polarization of T-helper cells towards the Th17 proinflammatory phenotype [16] was shown to be modulated by aldosterone. In a salt excess model, a similar, but macrophage-dependent, shift was observed [17] that led to disease aggravation in experimental autoimmune encephalitis (EAE) [18]. On the other hand, blocking the renin–angiotensin–aldosterone system by therapeutically blocking the angiotensin-converting enzyme can reverse paralytic EAE in mice [19]. In MS patients, higher salt intake correlated with an increased exacerbation rate in one study [20], while ensuing epidemiological studies could not yet confirm a role of salt intake in the clinical course of MS [21,22,23]. Therefore, the role of MR signaling in MS remains incompletely understood.

We set out to explore the potential functional impact and clinical usefulness of our earlier finding of decreased MR expression in MS patients. Since the net effect of MR signaling, like that through the GR, is diversely regulated on different levels (the extent of transcription, posttranscriptional modifications, ligand hormone binding, the presence of corepressors or -activators, cytosolic vs. nuclear location), we decided to measure the ultimate effect of these processes, i.e., the expression of downstream target genes activated by the administration of aldosterone. Thus, we used microarray technology to identify marker genes in PBMCs, confirmed one of them in a series of independent experiments and then measured its expression in MS patients and healthy controls.

As a patient sample, we chose untreated patients (many recruited at the time of initial diagnosis), patients on a traditional injectable disease-modifying treatment (DMT: interferon beta) and patients on a higher-efficacy DMT with a more precisely defined mode of action (fingolimod); for these groups, clinical data and blood samples could be collected in appropriate numbers through our outpatient clinic.

## 2. Results

### 2.1. MR Is Expressed in Major PBMC Subsets

Since our previous data on MR expression were derived from lysed whole blood samples, we first determined its expression in peripheral blood mononuclear cells (PBMCs) and their major subsets in ten healthy subjects. Therefore, we sorted PBMCs by magnetic beads into CD4+ (CD4 T lymphocytes), CD8+ (CD8 T lymphocytes) and CD19+ (B lymphocytes) fractions. GR was expressed in unsorted PBMCs and the defined cell fractions at almost equal levels (qRT-PCR, see Figure 1). The MR expression level was significantly, but only slightly, higher in CD8+ lymphocytes.

### 2.2. Ovarian Tumor Domain-Containing DUB 1 (OTUD1) Is a Marker for MR Signaling in PBMCs

Out of 13 potential marker genes that were identified from the microarray data, OTUD1 remained consistently and significantly upregulated in a series of 24 independent experiments upon ex vivo stimulation with aldosterone (Figure 2).

### 2.3. MR and OTUD1 Expression Can Reliably Be Measured in Lysed Whole Blood Samples

Since lysed whole blood is more easily acquired in clinical practice than isolated PBMCs, we measured the mRNA expression levels of MR and OTUD1 in both isolated PBMCs and lysed full blood in the first three patients. The results were comparable (see Figure 3) and lysed full blood was then used for all further measurements.

### 2.4. In Multiple Sclerosis, the Expression of Both MR and OTUD1 Is Altered

Having identified a marker gene for MR’s transcriptional activation, we analyzed its expression in 119 patients with MS and 42 healthy controls (see Table 1 for demographic and clinical characteristics). Most patients suffered from relapsing–remitting MS (RRMS). About one-third each were on no DMT, interferon beta or fingolimod, respectively. MR expression was significantly lower in MS patients than in healthy controls (Figure 4). The MR marker gene OTUD1 was also expressed in significantly lower amounts in MS patients than in healthy controls (Figure 4).

In order to explore a possible association of MR and marker gene expression with the clinical disability imposed by MS, we formed patient subgroups according to disability. Disability was measured using the Expanded Disability Status Scale (EDSS) score [24] (Figure 4, right-hand sections of graphs), a well-established tool for the clinical evaluation of patients with MS, both in research and routine settings. The expression of MR and OTUD1 was lower in all patient groups, with a trend towards but no consistent progressive decline. Intriguingly, the expression profile of OTUD1 largely paralleled that of MR expression.

To evaluate age- or sex-related effects, subgroups were formed as shown in Figure 5. In healthy controls, age and sex showed no significant impact on MR and OTUD1 expression. In patients with MS or CIS, MR expression was the lowest in younger and female patients. OTUD1 expression, on the other hand, was consistently lower in MS patients, independent of age and sex.

### 2.5. Disease-Modifying Treatments Differentially Affect MR and OTUD1 Expression

To next elucidate the influence of selected immunomodulatory treatments on MR signaling, we analyzed the expression levels according to DMT. Interestingly, while MR expression was lower in untreated patients than in healthy controls, interferon-β-treated patients’ MR expression was equal to that of the controls (Figure 6).

Patients treated with fingolimod showed strikingly low MR expression; this group differed significantly from all other groups. As shown in Figure 6, the expression of OTUD1 was decreased in all patients compared to the controls, which was also true for interferon-treated patients in spite of the normal expression of MR. Corresponding to the very low MR expression, the OTUD1 expression was lowest in fingolimod-treated patients, but to a far lesser degree. Age and sex did not influence these effects (see Figure 7).

## 3. Discussion

We could confirm our earlier finding that the MR expression in full blood is diminished in patients with MS [15]. To understand the functional consequence of this finding, we considered how the downstream signaling of MR would be changed in vivo. In order to answer this question, we first identified a gene regulated by the most important endogenous mineralocorticoid, aldosterone, in blood cells and found that its expression largely paralleled that of MR in patients with MS with varying disease states or therapeutic regimes.

### 3.1. Relevance of Detected MR Signaling and OTUD1 in the Context of Autoimmunity

Consistent with the role of mineralocorticoids and MR in the regulation of the immune response, OTUD1 has been reported or is strongly suggested to be involved in immunomodulatory function. OTUD1 is a member of the family of deubiquitinating enzymes (DUBs), which cleave ubiquitin bonds and are therefore essential for protein degradation in various cellular functions. DUBs influence interferon production in T lymphocytes, indirectly contributing to the negative regulation of the innate immune response [25,26]. Dysfunction of OTUD1 can contribute to excessive interferon production and thus autoimmunity; this is supported by the association of loss-of-function mutations in OTUD1 with autoimmune diseases (lupus erythematosus, rheumatoid arthritis or Hashimoto’s thyroiditis) [27]. In inflammatory bowel disease, the mucosa was found to express lower levels of OTUD1 [28]. Recently, a contribution to the severity of T-cell-mediated acute graft-versus-host disease in mice was shown via the OTUD1-dependant accumulation of Notch2-ICD, inducing a shift towards the pro-inflammatory Th17 phenotype [29]. Because of its possible implications in the pathogenesis of inflammatory and oncological diseases, OTUD1 inhibitors are being developed [30]. We are the first to report the regulation of this gene by aldosterone administration and the potential role in the pathogenesis of autoimmunity and/or inflammation in MS.

### 3.2. The Search for MR Marker Genes Relevant In Vivo

As our goal was to find MR downstream marker genes relevant in vivo, we decided to work under physiological conditions as much as possible. These included incubation in autologous plasma, the use of natural cortisol and aldosterone rather than synthetic ligands with higher receptor affinity and transactivation potency and the administration of physiological rather than pharmacological ligand concentrations. We decided to measure the expression levels in order to take most known and unknown factors orchestrating MR activity in vivo into account. Such factors include the regulation of MR transcription or translation on many levels, posttranscriptional modification, competition with other hormone receptors and the presence of corepressors or -activators [31,32]. Moreover, to ensure applicability and transferability, we did not block corticoids with their respective pharmacological antagonists, since cross-activation (or -repression) of receptors occurs in such co-incubation experiments [33]. Admittedly, the achieved transferability comes at the expense of a mechanistical understanding of the detailed interplay of single factors. One obvious point is that, in addition to the aldosterone added experimentally, other steroids present in the autologous plasma may have contributed to receptor (and ensuing gene) activation. Taking into account the higher binding affinity of MR (Kd ≈ 0.1–1 nM) for corticosteroids compared to GR (Kd ≈ 10.25 nM), endogenous cortisol is expected to saturate MR, especially since blood cells express little or no 11beta hydroxysterol dehydrogenase 2 (11β-HSD2) [34], which prevents this effect intracellularly by metabolizing cortisol to cortisone. However, 11β-HSD2 activity is not the only mechanism determining tissue’s susceptibility to mineralocorticoid effects [31,32]. In particular, the slower dissociation of aldosterone from the MR can be expected to override cortisol’s effects in the experimental setup used. In order to limit the effect of cortisol, we focused our MicroArray analysis and selection of candidates for confirmation experiments more specifically on aldosterone-regulated genes by excluding those with strong regulation by cortisol. Genes known to be regulated by both MR and GR (FKBP5 [35], Sgk1 [36]) and candidates with biologically plausible regulation by cortisol (such as cytokines and chemokines) therefore did not reach the top ranks of our putative target gene list. We may thus have missed target genes on which MR has a more pronounced effect. Lastly, we reduced the risk of type I errors caused by multiple testing using the relatively conservative closed testing procedure.

Taking into account these restrictions, and although the effects of aldosterone on the induction of OTUD1 were relatively small, in vitro induction was consistent and we therefore claim that this gene can serve as a marker gene of MR signaling in peripheral blood cells. The results in our in vivo sample support this notion since MR and OTUD1 expression profiles were similar in most group comparisons; at the same time, the differences between the MR and OTUD1 expression levels in patients treated with interferon ß and fingolimod, respectively, suggest redundant regulation of OTUD1 expression, which we cannot further explain at this stage.

### 3.3. Role of MR Signaling in MS

As briefly described in the Introduction, evidence has accumulated regarding the role of MR in modulating immune function and inflammation in EAE and potentially in MS. These changes were mostly observed upon aldosterone or excessive salt administration. A recent study measured the levels of proinflammatory cytokine and disability in mice with EAE and found that eplerenone, a specific MR antagonist, dampened not only the disease course and disability but also cytokine levels. In the spinal cord gray matter, an imbalance of GR and MR was observed. The lower expression of MR in EAE was reversed by the therapeutic administration of eplerenone [37]. These findings are in line with our observations and underline the possible role of MR in MS; our study contributes to the sparse human data obtained so far.

In contrast to healthy controls, patients’ MR expression levels were especially low in female and younger patients (see Figure 5). Data on the expression levels of MR and OTUD1 in peripheral blood are missing and, hence, there is no information on age- or sex-related changes. Possibly, sex-specific aspects of HPA axis regulation could contribute to the more pronounced effects in female patients. Rodent studies indicate that females present with a more pulsatile basal HPA output that reaches higher peak doses of corticosterone upon equal amounts of ACTH, indicating higher sensitivity to stimulation. The stress response in females is associated with higher blood concentrations of corticosterone earlier and for a longer duration, underlining that the female HPA axis appears more dynamic. Gonadectomy in adult rats leads to heightened ACTH and corticosterone release following stress, indicating the inhibiting effects of androgens on the HPA axis, a potentially protective factor for male patients [38].

Looking at the differences in age, many of our patients were newly diagnosed, and the effect of MS itself on MR expression was most pronounced and not yet confounded by treatment. With an older age, a decrease in MR expression in the hippocampus and high peripheral cortisol levels lead to reduced negative HPA feedback, resulting in chronically high concentrations of ACTH and cortisol [39]. Since this would most likely result in lower peripheral MR levels with increasing age, we claim that in our rather young sample, MS-specific changes predominate over age-dependent effects on MR expression.

Changes in the expression of OTUD1, on the other hand, were consistent for all treatment groups and independent of age or sex. When further analyzing the patient subgroups according to their EDSS scores, OTUD1 expression largely paralleled MR expression; with increasing disability, there is a trend for the expression levels of MR and its downstream target OTUD1 to decrease, suggesting at least a potential impact of MR signaling on disability progression (or vice versa). In order to confirm this, it will be helpful to study larger groups, specifically including patients with higher disability scores. As stated above, the finding of this largely parallel expression pattern across the disability subgroups supports the designation of OTUD1 as an MR-regulated gene.

The observations in treated patients merit further discussion. Patients on interferon β expressed MR in comparable amounts to healthy controls, which may add yet another aspect to the diverse known mechanisms involved in its therapeutic efficacy [40]. The possible regulation of MR expression by interferon ß has not yet been described but seems possible, as the promoter region of the GR contains a response element for interferon [41]. Remarkably, the same patients had low OTUD1 expression despite normal MR. This indicates redundant regulation on additional levels but could also be understood as a sign of the sustained dysregulation of the immune system in favor of autoimmunity, as a loss of function in the OTUD1 gene was seen to be associated with autoimmune diseases [27].

Patients on fingolimod had remarkably low expression of MR. The simplest explanation would be that MR is mainly expressed in lymphocytes, whose peripheral blood count is drastically reduced by fingolimod. We doubt this hypothesis, as we found that MR expression differed only slightly among magnetically selected lymphocyte subsets; in addition, the expression was similar in unsorted peripheral blood mononuclear cells and full blood, i.e., essentially neutrophils, whose count is unaffected by fingolimod. At this stage, we cannot provide a mechanistic explanation for the very low MR expression.

### 3.4. Future Aims and Perspectives

In order to confirm the pathophysiological role of our findings, future work should confirm our findings at the protein level and further explore the mechanisms of MR signaling in the identified marker gene and potential co-regulatory effects. This could be supplemented by steroid binding assays, the identification of corepressors and -stimulators and the proof of a functional protein product. To explain the very low MR expression in patients treated with fingolimod, single-cell RNA analysis would be helpful.

Moreover, the role of MR signaling in immune cells involved in MS pathogenesis and the impact of HPA axis regulation on systemic MR signaling merit further research. In MS specifically, we propose collecting longitudinal data from larger groups on various disease-modifying treatments to corroborate the role of MR signaling in the clinical course; having identified an MR marker gene in peripheral blood will facilitate such an analysis. Eventually, MR and OTUD1 could contribute to the monitoring of patients and to personalized therapy, an identified need in MS treatment [42]. To this end, the available clinical, electrophysiology and magnetic resonance imaging techniques could be supplemented by body fluid biomarkers. Among those proposed (and reviewed in [43]), some have failed (BDNF, leptin, copeptin) and others are still being evaluated (e.g., CSF-CXCL13, CSF-OPN, CSF-CHI3L1, circular and micro-RNAs), while, currently, two appear most promising: glial fibrillary acid protein (GFAP) and neurofilament light chain (NfL). The former is an astrocytic protein and may, for example, distinguish between relapse and remission in RRMS. The latter (NfL) is an axonal structural protein and serves as a marker for neuronal damage in CNS diseases. In MS, its (CSF or serum) concentration is associated with the conversion risk from CIS to MS, relapse activity, MRI dynamics and disability progression [43,44]. More comprehensive analyses have shown that NfL, in combination with ten additional peptides, could predict disability worsening [45]. Serum GFAP, serum Nfl and the combination of both is being validated to distinguish between relapse/progression and remission/stable disease in RRMS and PPMS [46]. While it is becoming clear that a panel of soluble markers will probably be superior to a single metabolite, these potential biomarkers must also prove their clinical usefulness. For example, the biochemical effects of treatment response could be detected earlier than in the current clinical and imaging protocols, allowing for timely adjustments of treatment. We suggest that MR and OTUD1, given their changes upon specific treatments, are candidates for the evolving set of biomarkers in MS. Clearly, confirmation in a separate sample and longitudinal observation are required to establish their role in this context and, ultimately, prospective clinical validation.

## 4. Conclusions

With OTUD1, we could identify and validate a marker gene of MR signaling in peripheral blood in ex vivo conditions, although its expression appears to be regulated by additional factors. MR expression and MR signaling were altered on the transcriptional level in MS compared to controls. Since OTUD1 influences cytokine production and has an impact on the evolution and maintenance of experimental autoimmunity, we postulate an MR-mediated contribution to the course of MS. This appears further supported by the observed alterations of the expression levels in patients receiving DMT. This claim must be further investigated in a larger, longitudinal cohort of MS patients.

## 5. Materials and Methods

### 5.1. Subjects

For cell culture experiments with peripheral blood mononuclear cells (PBMCs), peripheral blood was collected between 06:00 and 08:00 in the morning from healthy volunteers on a single occasion with their informed consent, using BD Vacutainer^®^ CPT™ Mononuclear Cell Preparation Tubes (BD Biosciences, Franklin Lakes, NJ, USA). Volunteers were considered healthy if neither taking any regular medication (and no as-needed medication for at least one week) nor suffering from any other known medical condition. We included 12 male and 12 female volunteers, aged 31.7 ± 8.7 years.

For the analysis of gene expression in MS patients vs. controls, we recruited 119 consecutive patients with MS or clinically isolated syndrome (CIS, presumed first episode of MS) from our clinic on the occasion of routine visits, after clinical interview and examination with the determination of the Expanded Disability Status Scale (EDSS) [24]. We included patients either not yet on DMT or using interferon beta or fingolimod during the past 12 months. We excluded patients suffering from any other severe disease or having received systemic glucocorticosteroids within the last eight weeks. During venipuncture for routine laboratory investigations, we collected an additional sample into Tempus^®^ Blood RNA Tubes (Applied Biosystems, Waltham, MA, USA); in the first three patients, an additional sample was collected for PBMC preparation. We chose healthy controls to largely match the patients with respect to age and sex (Table 1).

The collection and processing of subjects’ blood samples was approved by the Ethics Committee of the Medical Faculty, University of Leipzig (210/13-ff and 265-12-15072013), and patients and controls gave their informed consent. All methods were performed in accordance with the relevant guidelines and regulations.

### 5.2. Clinical Assessment of Patients’ Disability (EDSS)

We measured the extent of neurological impairment caused by MS using the Expanded Disability Status Scale (EDSS) [24]. It is based on clinical findings in the neurological examination and is well established to monitor disease severity and progression. Scores range from 0 (no deficit) to 10 (death due to MS) with 0.5 increments. We formed groups according to functional impairment: scores from 0 to 1.5 indicate clinical findings upon examination without relevant impairment, 2.0–3.0 denote mild impairment and 3.5–4.5 denote impairment with preserved walking ability. Scores 5.0 and higher are associated with impairments in walking ability and a growing dependency in daily activities.

### 5.3. Cell Culture

From the CPT™ tubes, PBMCs were isolated according to the manufacturer’s instructions and re-suspended in autologous plasma. Only samples with a maximum of 5% of dead cells (as assessed by their ability to extrude 0.4% trypan blue on phase contrast microscopy) and a minimum amount of 1 × 10^6^ cells/mL were used. Where indicated (see Figure 1), PBMCs were magnetically sorted into CD4+, CD8+ and CD19+ lymphocytes, using the EasySep™ Cell Separation Kit from STEMCELL Technologies (Vancouver, BC, Canada, following the manufacturer’s instructions. Cells were incubated at room temperature in autologous plasma with aldosterone, cortisol or vehicle (as indicated per experiment) for three hours.

Concentrations of steroid hormones were chosen within a physiological range [47] and supported by the results of a dose-finding experiment that demonstrated an optimal increase in GR target gene mRNA expression at a cortisol concentration of 2.76 µmol/L (see Appendix A). These figures allowed us, by using cortisol at 2.76 µmol/L and aldosterone at 2.77 nmol/L, to preserve the physiological cortisol-to-aldosterone ratio of approximately 1000:1 (reported ratios range from 100:1 to 2000:1 [31,47,48]). The aldosterone concentration for cell culture experiments was validated after the identification of OTUD1 as a marker gene by conducting a larger dose-finding experiment (Appendix A). A linear dose dependency was shown at concentrations ranging from 0.00277 nmol/L to 2.77 nmol/L. Higher dosages did not increase the transcription rate further, suggesting that the system was saturated. The dosage of 2.77 nmol/L was therefore used in all further experiments. Vehicle samples contained the highest concentration of vehicle (ethanol) used in each experiment to achieve the required hormone concentrations and ranged between 4.8 mmol/L and 17.13 mmol/L.

### 5.4. RNA Isolation and cDNA Synthesis

RNA was extracted from PBMCs using the RNeasy Kit^®^ (Qiagen, Hilden, Germany) according to the manufacturer’s instructions. Tempus™ samples were processed using the Tempus^TM^ Spin RNA Isolation Reagent Kit (Applied Biosystems^TM^, Waltham, MA, USA). The entirety of the extracted RNA was reverse-transcribed using random hexamer primers and the High-Capacity cDNA Reverse Transcription Kit with RNase Inhibitor (Applied Biosystems^TM^, Waltham, MA, USA) and stored at −80 °C.

### 5.5. Gene Expression Analysis (Microarray)

Two Illumina HT12 Beadchips were used to generate a list of potential MR marker genes that were upregulated upon MR stimulation by aldosterone. The experimental details as well as the strategy used to search for MR-stimulated transcripts are given in the Appendix A.

### 5.6. Quantitative Real-Time PCR

Oligonucleotide primers were designed to span intron sequences, using the Primer 3 Software, release 4.1.0 [49,50]. Agarose gel electrophoresis verified the correct amplicon size and primers with efficacy of 90–110% compared to GAPDH were accepted. The sequences used are shown in Appendix A To quantify the transcript levels, we performed real-time polymerase chain reactions using Platinum SYBR Green^®^ qPCR Supermix (Life Technology), combining 6.25 µL SYBR green^®^, cDNA (300 ng from full blood samples, 600 ng from PBMC samples), 0.6 mM each of the forward and reverse primers and 100 nM ROX (5-carboxy-X-rhodamine passive reference dye). To confirm the presence of a single amplicon product, melting curves were recorded. Amplification was performed in a two-step cycling protocol using MX Pro 3000P by Stratagene (Agilent Technologies, Santa Clara, CA, USA). The protocol of RT-PCR used was as follows: 2 min 50 °C, 2 min 95 °C, followed by 50 cycles of 15 s at 95 °C and 30 s at 60 °C.

To determine the relative expression levels of the target genes in cell culture samples, data were normalized to GAPDH (ΔcT) and ΔΔcT values were formed by subtracting the vehicle values from the remaining groups [51]. In the whole blood samples, the ΔcT values were calculated with reference to the GAPDH expression levels. The graphs are presented as ΔcT, $2^{-∆∆cT}$ and $2^{-∆cT}$ values, as indicated in the figure legends.

### 5.7. Statistics

For cell culture samples, the ΔcT values of each sample were used for statistical analysis versus the vehicle. A closed testing procedure [52] was performed after the first six samples to correct for type I errors, and genes showing relevant differences were then compared using a two-tailed Student’s *t*-test assuming equal variance. This was also applied for the ΔcT values of magnetically sorted cells, which were compared to unsorted cells’ values. In the whole blood samples, the $2^{-∆cT}$ values were used for statistical analysis. One-way ANOVA followed by Tukey’s post hoc test was used to compare multiple groups. Findings with *p* < 0.05 were considered statistically significant.

## Figures and Tables

**Figure 1 ijms-25-08883-f001:**
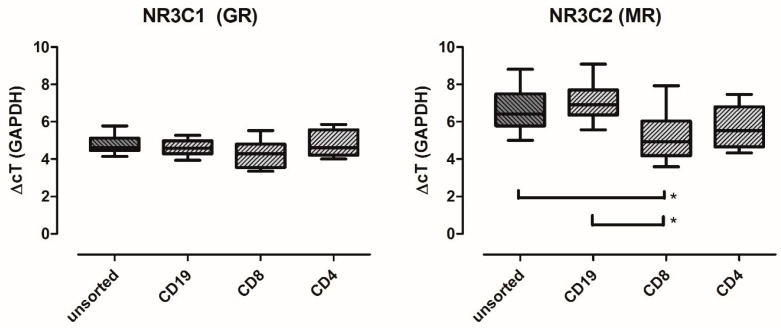
Comparison of GR and MR expression levels in blood samples of ten healthy subjects. Shown are the expression levels in unsorted leukocytes (dark grey shading) and magnetically sorted subsets as indicated on the *x*-axis. Asterisks mark statistical significance in one-way ANOVA and Tukey’s post hoc test (*p* < 0.05).

**Figure 2 ijms-25-08883-f002:**
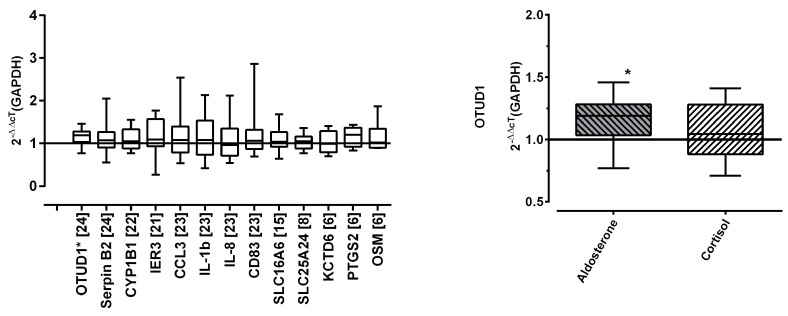
Validation of downstream MR target. Left: Putative target genes’ expression levels upon stimulation with aldosterone (dosage 2.77 nmol/L). Number of samples as indicated in brackets. Boxplots mark median and two-sided 25% quartiles; whiskers indicate minimum to maximum values. Right: Expression profiles of OTUD1 in PBMCs after stimulation with aldosterone (dark grey bar, identical samples as in panel A, *n* = 24, dosage 2.77 nmol/L) and cortisol (light grey bar, *n* = 12, dosage 2.76 µmol/L). Boxplots mark median and two-sided 25% quartiles; whiskers indicate minimum to maximum values. Asterisks assign statistical significance in paired *t*-test of ΔcT values against vehicle (*p* < 0.05). *p*-value for OTUD1 was <0.001 (vehicle vs. aldosterone).

**Figure 3 ijms-25-08883-f003:**
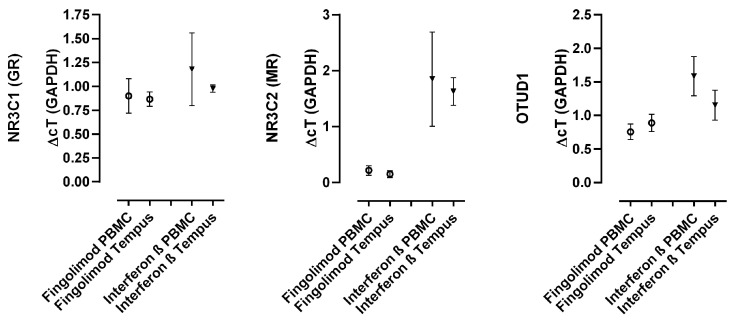
Comparison of expression levels with samples derived from full blood (Tempus^®^ tubes) and PBMCs in patients with multiple sclerosis (*n* = 3; DMT as indicated on *x*-axis). Gene expression levels normalized to GAPDH of GR, MR and OTUD1 as indicated on the *y*-axis. Data are shown as the mean and standard error of the mean (SEM).

**Figure 4 ijms-25-08883-f004:**
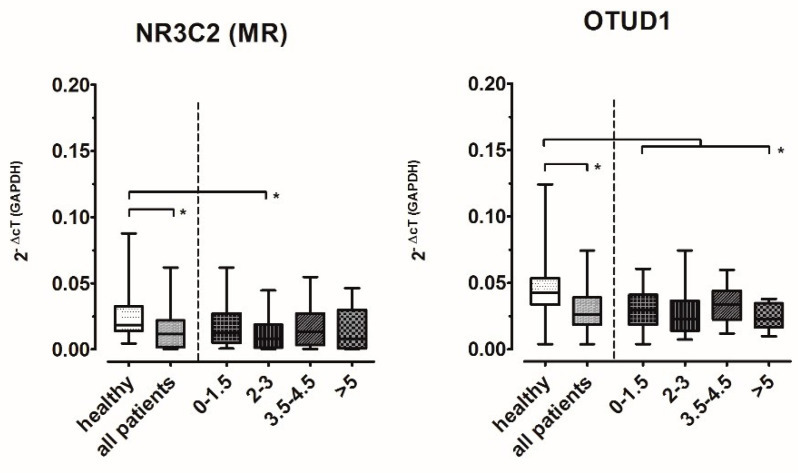
Gene expression levels of MR and OTUD1 in MS patients and healthy controls. In each panel, the left section shows healthy controls (*n* = 42) versus all patients (*n* = 119). Boxplots mark median and two-sided 25% quartiles; whiskers indicate minimum to maximum values. *t*-test showed significant differences (*p* < 0.05). The right section shows expression data in patient groups based on EDSS ranges as indicated. ANOVA was used to test for overall significance; brackets and asterisks mark statistically significant differences identified by Tukey post hoc testing where ANOVA was significant (*p* < 0.05). Subject counts are shown in Table 1.

**Figure 5 ijms-25-08883-f005:**
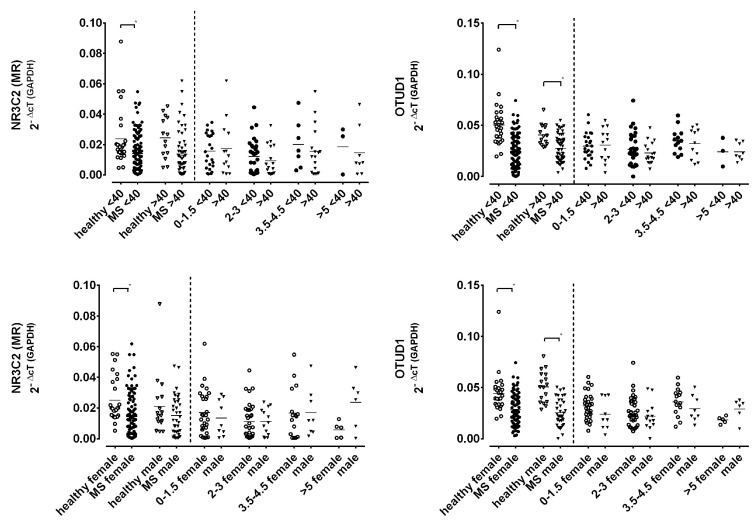
Gene expression levels of MR and OTUD1 in patients with MS and healthy controls with respect to age (upper graphs) and sex (lower graphs). Left sections of graphs show comparison of all patients and healthy controls according to age (<40 years and >40 years) and sex (male, female). Right sections of graphs compare patients’ expression levels in EDSS subgroups for age and sex. Data are shown as scatter dot plots with means. Asterisks mark significant differences in Tukey post hoc test of one-way ANOVA for the left sections of the graphs. EDSS subgroups did not undergo statistical testing.

**Figure 6 ijms-25-08883-f006:**
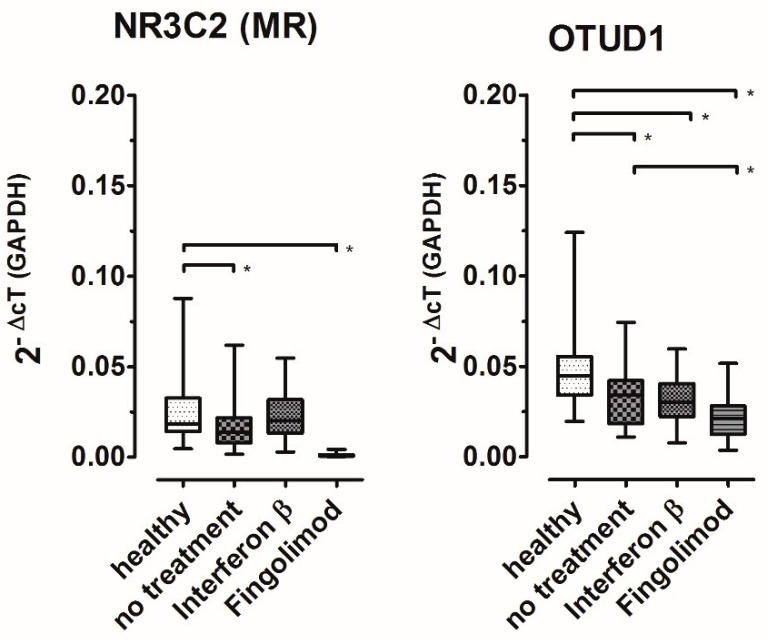
Gene expression levels of MR and OTUD1 according to therapeutic regime. Boxplots mark median and two-sided 25% quartiles; whiskers indicate minimum to maximum values. Brackets and asterisks mark statistically significant results in ANOVA with Tukey post hoc testing (*p* < 0.05). Subject counts are shown in Table 1.

**Figure 7 ijms-25-08883-f007:**
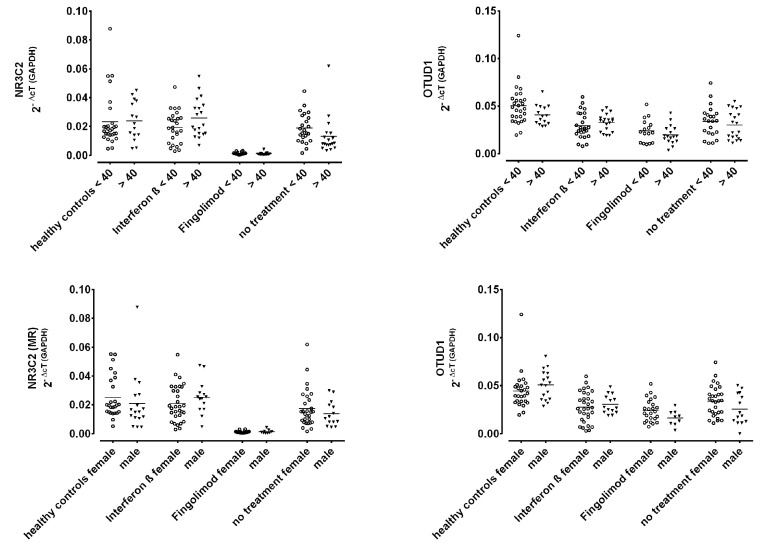
Gene expression levels of MR and OTUD1 in patients with MS according to therapeutic regime and healthy controls with respect to age (upper graphs) and sex (lower graphs). Data are shown as scatter dot plots with means. There were no significant differences in Student’s *t*-test when comparing subgroups for age and sex, respectively.

**Table 1 ijms-25-08883-t001:** Patients‘ and healthy controls‘ demographic and clinical data.

	Healthy Controls	MS Patients	EDSS Groups					
			0–1.5	2–3		3.5–4.5		>5
*n* (%)	42	119	37 (31.1)	47 (39.5)		24 (20.2)		11 (9.2)
Age [years]: mean (SD)]	33.6 (10.2)	38.9 (10.4)	35.0 (9.9)	36.7 (8.9)		45.8 (10.6)		46.0 (7.2)
Female: *n* (%)	25 (59.5)	86 (72.3)	28 (75.7)	37 (78.7)		16 (66.7)		5 (45.5)
Disease duration [years]: mean (SD)		6.4 (7.0)	2.9 (3.3)	4.8 (5.1)		12.9 (9.1)		11.2 (7.5)
Disease course [*n*] (CIS/RRMS/SPMS)		(14/99/6)	(8/29/0)	(6/41/0)		(0/23/1)		(0/6/5)
Treatment: *n* (%)								
no treatment		42 (35.3)	19 (51.4)	17 (36.2)		2 (8.3)		4 (36.4)
interferon β		44 (37.0)	11 (29.7)	14 (29.8)		15 (62.5)		4 (36.4)
fingolimod		33 (27.7)	7 (18.9)	16 (34.0)		7 (29.2)		3 (27.3)
	**Healthy controls**	**MS patients**	**Treatment**					
			no treatment		interferon β		fingolimod	
*n* (%)	42	119	42 (35.3)		44 (37.0)		33 (27.7)	
Age [years]: mean (SD)	33.6 (10.2)	38.9 (10.4)	37.6 (10.4)		38.5 (10.8)		41.1 (9.3)	
Female: *n* (%)	25 (59.5)	86 (72.3)	28 (66.7)		31 (70.5)		24 (72.7)	
Disease duration [years]: mean (SD)		6.4 (7.0)	3.6 (5.4)		7.5 (7.4)		8.6 (7.3)	
Disease course [*n*]: (CIS/RRMS/SPMS)		(14/99/6)	(13/27/3)		(1/42/1)		(0/30/2)	

Patients were recruited consecutively from the clinic, and controls showed a similar distribution in terms of age and sex. In patients, disease-modifying treatments had to be prescribed for at least 12 months, without glucocorticoids as relapse therapy, within eight weeks prior to the drawing of the blood samples.

## Data Availability

The discussed data (non-normalized and quantile data with subtracted background) are available at NCBI’s Gene Expression Omnibus [33] through GEO Series accession number GSE162695 (https://www.ncbi.nlm.nih.gov/geo/query/acc.cgi?acc=GSE162695). Restrictions apply to the availability of some data generated or analyzed during this study. The corresponding author will, on request, provide details about the restrictions and any conditions under which access to the data may be provided.

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
