# Peer review of "Mineralocorticoid Receptor Signaling in Peripheral Blood Cells in Patients with Multiple Sclerosis"

_ijms, 2024, doi:10.3390/ijms25168883_

Round 1

Reviewer 1 Report

Comments and Suggestions for Authors

The authors are encouraged to add some information on the specific conditions used for cell culture, as well as provide a clear explanation for why they chose the particular concentration of aldosterone for stimulation.

The study should provide a more explicit description of the sample size and selection procedures for both multiple sclerosis patients and healthy controls in order to enhance the reproducibility of the research.

The manuscript should include information on how age and sex affect MR and OTUD1 expression, considering any potential factors that could influence the results.

The discussion section should be broadened to provide a more comprehensive analysis of the potential mechanisms through which MR signaling could impact MS development and treatment response.

The authors should address the limitations of this work, specifically the cross-sectional methodology and the absence of longitudinal data. Longitudinal data would offer valuable insights into the temporal dynamics of MR and OTUD1 expression in connection to disease activity and treatment effectiveness.

Additionally, the manuscript's quality should be improved by include information on how these findings compare to and supplement current biomarkers in MS, as well as the practical considerations for integrating these elements in clinical practice.

Comments on the Quality of English Language

The English quality of the manuscript requires minor editing

Author Response

(Line numbers refer to the “Marked” version of the revised manuscript)

Comment: The authors are encouraged to add some information on the specific conditions used for cell culture, as well as provide a clear explanation for why they chose the particular concentration of aldosterone for stimulation.

Response: We moved a more detailed description of the cell culture from the supplementary section to the main text (now lines 482-511). The figure (S2) showing the result of the dose finding experiments, resulting in the chosen aldosterone concentration, is included in the supplementary materials and, at the discretion of the editor, can be moved to the main text.Comment: The study should provide a more explicit description of the sample size and selection procedures for both multiple sclerosis patients and healthy controls in order to enhance the reproducibility of the research.

Response: We added the requested information to the methods section (now lines 438-458). In short, consecutive patients from the clinic were included to the study. Selection criteria were correct diagnosis, current disease-modifying treatment (none, interferon beta or fingolimod), absence of other severe comorbidities, absence of steroid treatment within eight weeks.  We selected healthy volunteer controls to largely match patients.

Comment: The manuscript should include information on how age and sex affect MR and OTUD1 expression, considering any potential factors that could influence the results.

Response: In addition to our own results (fig. 5) we added some literature on HPA regulation according to sex and age, in the discussion section (lines 317-339). Data for OTUD1 on this question is missing.

Comment: The discussion section should be broadened to provide a more comprehensive analysis of the potential mechanisms through which MR signaling could impact MS development and treatment response.

Response: We tried to avoid becoming too speculative in our first draft of the work, but we added some thoughts about this topic (now lines 304-315); we also added a reference to a recent original contribution (Alvarez Quintero et al., 2024).

Comment: The authors should address the limitations of this work, specifically the cross-sectional methodology and the absence of longitudinal data. Longitudinal data would offer valuable insights into the temporal dynamics of MR and OTUD1 expression in connection to disease activity and treatment effectiveness.

Response: We acknowledge that all these limitations apply, and while we had addressed them, the limitations section was broadened accordingly (now lines 349-351, 377-384, 387-389, and 415-418).

Comment: Additionally, the manuscript's quality should be improved by include information on how these findings compare to and supplement current biomarkers in MS, as well as the practical considerations for integrating these elements in clinical practice.

Response: With the lack of confirmatory analyses in mind, we added some outlook on how MR and OTUD might supplement current (imaging, electrophysiology and body fluid) biomarkers (lines 393-413). We envision that MR or/and OTUD1 could supplement biomarkers of early biochemical response to immunomodulatory treatment in order to identify potential future treatment failures.

Reviewer 2 Report

Comments and Suggestions for Authors

This paper attempt to find Mineralocorticoid receptor (MR) downstream marker genes relevant in vivo. The authors previously found that MR expression in full blood is diminished in patients with multiple sclerosis (L167). Ex vivo experiments in gene expression to aldosterone provided a candidate, OTUD1. The authors investigated the expression level of the gene in MS patients with several data stratifications. As a result, the authors claim that although OTUD1’s expression appears to be regulated by additional factors, OTUD1 is a marker gene of MR signaling. No deficiencies in experimental procedures or ethics were found. However, this reviewer believes that the questions set out below need to be answered with regard to statistical interpretation. In addition, this reviewer believes that basically this report is in line with the subject matter of this Special issue, but with a focus on OTUD1 in the clinical samples. (That is meaning it is difficult to prepare the same samples by experiment.) Therefore, this reviewer would recommend submission to a more specialized journal for medicine rather than this journal dealing with molecular science.

1) (Fig 6 and L243-249)

The expression level of MR varies greatly depending on the drug used. Are there any differences in the pathology of the interferon b group and the Fingolimod group during the actual course of treatment? In the graphs from Fig. 6 onwards, stratification by EDSS is not discussed.

2) (Fig 4 and L167-168)

As discussed in 1), to what extent does the contribution of medication affect the assessment in, for example, Fig. 4?

3) (Fig 7 and L230-233)

This reviewer thinks that this part is not explained well enough. Please explain more.

4) the whole document and its analysis

The authors used Tukey post-hoc test for evaluation, but the changes in disease indicated by MR and OTUD1 presented here are too small to be used as reliable markers. This reviewer believes that the authors should evaluate the statistical power, including false negatives and false positives.

typo

L124: “EDSS”, spell out as expanded disability status score (?)

L129: 24 -> [24]

Author Response

(Line numbers refer to the “Marked” version of the revised manuscript)

Comment: 1) (Fig 6 and L243-249)

The expression level of MR varies greatly depending on the drug used. Are there any differences in the pathology of the interferon b group and the Fingolimod group during the actual course of treatment? In the graphs from Fig. 6 onwards, stratification by EDSS is not discussed.

Response: In MS, differences in pathology have been described histologically, but cannot be detected in vivo (with the rare exception of patients whose diagnosis is made on the initial suspicion of brain tumor and have therefore undergone biopsy). No such patients happened to be included in our sample.

When we stratified for the respective treatment groups, we resigned from an additional stratification according to EDSS as the resulting groups would have been too small to allow comparisons.

In general, EDSS rates neurological disability due to MS, where 0-1.5 reflects at most abnormal findings on neurological examination without impairment, 2-3 mild impairment with preserved activities of daily living, 3.5-4.5 moderate disability with preserved walking ability, and from 5.0 on, with additional significant walking impairment. We included this in the methods section (lines 165-168 and 471-480) to account for readers not so familiar with this clinical score.

Comment: 2) (Fig 4 and L167-168)

As discussed in 1), to what extent does the contribution of medication affect the assessment in, for example, Fig. 4?

Response: We agree that the proportion of patients taking fingolimod is indeed largest in the EDSS 2-3 group. This may have enhanced the effect of stratifying the sample according to  EDSS, although the difference is not large. It will be important to analyze a larger sample in subgroups to verify or falsify this and potential additional confounders.

Comment: 3) (Fig 7 and L230-233)

This reviewer thinks that this part is not explained well enough. Please explain more.

Response: We acknowledge that the text in the former L 230-233 refers to fig 5 rather than fig 7, which may cause some confusion to the reader. We have therefore added a reference to the correct figure in the text (line 317).

Comment: 4) the whole document and its analysis

The authors used Tukey post-hoc test for evaluation, but the changes in disease indicated by MR and OTUD1 presented here are too small to be used as reliable markers. This reviewer believes that the authors should evaluate the statistical power, including false negatives and false positives.

Response: We agree (and have stated in the discussion) that the differences described are noticeable enough and biologically plausible, but not sufficiently validated clinically. We are of the opinion that the sample size, cross-sectional design and monocentric setting would not be sufficient to validate its clinical application. In this situation, a detailed biometric analysis may suggest a higher grade of validity than we would like to propose. Rather, we have outlined an approach to validate the result in a separate, larger sample, and stressed this outlook more in the discussion (lines 387-389, 413-418) and conclusion (lines 428/429).

Comment: typo

L124: “EDSS”, spell out as expanded disability status score (?)

L129: 24 -> [24]

Response: We thank the reviewer for careful reading and have corrected these errors.

Round 2

Reviewer 1 Report

Comments and Suggestions for Authors

The authors completed the manuscript. I consider it proper to be accepted.